# Mathematical Modeling and Software Tools for Breeding Value Estimation Based on Phenotypic, Pedigree and Genomic Information of Holstein Friesian Cattle in Serbia

**DOI:** 10.3390/ani13040597

**Published:** 2023-02-08

**Authors:** Ljuba Štrbac, Doni Pracner, Momčilo Šaran, Dobrila Janković, Snežana Trivunović, Mirko Ivković, Laslo Tarjan, Nebojša Dedović

**Affiliations:** 1Faculty of Agriculture, University of Novi Sad, 21000 Novi Sad, Serbia; 2Faculty of Science, University of Novi Sad, 21000 Novi Sad, Serbia; 3Faculty of Technical Science, University of Novi Sad, 21000 Novi Sad, Serbia

**Keywords:** milk production traits, cattle, SNP, accuracy of breeding values, genomic selection

## Abstract

**Simple Summary:**

This research aims to evaluate the possibility of introducing modern methods for the selection of dairy cattle in Serbia. A total of eight models were applied to estimate the genetic values and their accuracy based on the data of milk yield, milk fat yield, milk fat content, milk protein yield and milk protein content. A total of 6041 animals were included, and of them, 2565 had data for milk production traits. Two cases were studied, the first when only the first lactations were observed, and the second when all lactations were observed using a repeatability model. For the formation of a genomic relationship matrix, 58 K SNP data were used for a total of 1491 cows. Higher accuracy was obtained using models with repeated measurements. Multivariate analysis with repeated measurements showed the best results. An ssGBLUP methodology showed promising results. Expanding the genotyped population is necessary to achieve better accuracy.

**Abstract:**

In this paper, six univariate and two multivariate best linear unbiased prediction (BLUP) models were tested for the estimation of breeding values (BV) in Holstein Friesian cattle in Serbia. Two univariate models were formed using the numerator relationship matrix (NRM), four using the genomic relationship matrix (GRM). Multivariate models contained only an NRM. Two cases were studied, the first when only first lactations were observed, and the second when all lactations were observed using a repeatability model. A total of 6041 animals were included, and of them, 2565 had data on milk yield (MY), milk fat yield (FY), milk fat content (FC), milk protein yield (PY) and milk protein content (PC). Finally, out of those 2565 cows, 1491 were genotyped. A higher accuracy of BV was obtained when using a combination of NRM and GRM compared to NRM alone in univariate analysis, while multivariate analysis with repeated measures gave the highest accuracy with all 6041 animals. When only genotyped animals were observed, the highest accuracy of the estimated BV was calculated by the ssGBLUPp model, and the lowest by the univariate BLUP model. In conclusion, the current breeding programs in Serbia should be changed to use multivariate analysis with repeated measurements until the optimal size of the reference population, which must include genotyping data on both bulls and cows, is reached.

## 1. Introduction

Animal breeding, by studying variability, primarily genetic, and using different statistical methods, aims to select parents of the next generations that are characterized by high production capacity, appropriate functional and conformational characteristics, and the ability to pass them on to offspring. Selection programs, when setting the criteria for the selection of parents, rely on the statistical analysis of large databases made up of data on the origin and the phenotype measurements of the desired traits, most often by the method of the best linear unbiased prediction (BLUP), which is used to estimate the breeding values of candidates for selection. In the last few years, traditional methods of selection, based on quantitative genetics, have increasingly been complemented by molecular genetic analyses. Molecular genetics has enabled the direct analysis of animal genomes, and for this purpose polymerase chain reaction (PCR), analysis of genetic markers, DNA sequencing, etc., are used. Several types of genetic markers have been developed, of which single nucleotide polymorphism (SNP) is the most widely used today. By providing insight into the structure and function of the genome, SNPs have become powerful tools for improving the selection of all types of domestic animals. On the basis of this knowledge, genomic selection has been recently developed, a new method that enables greater genetic progress compared to previous forms of selection. Genomic selection is based on applying statistical models to a set of fully phenotyped and genotyped individuals, and then using the applied model to predict the genetic value of unmeasured individuals [1]. The first official genomic evaluations in the United States were published in 2009 for the Holstein, Jersey, and Brown Swiss breeds. The incorporation of DNA-marker technology and genomics into traditional methods for evaluating animal breeding values has doubled the rate of genetic progress for economically important traits, reduced the generation interval, increased selection accuracy, reduced costs relative to progenic testing and enabled the identification of recessive lethal alleles [2]. The core data set, which makes such prediction possible, is the reference data set consisting of animals for which three sources of information are available: SNP genotypes, phenotypes and pedigree [3].

Modern breeding programs that include molecular methods for data collection on animals generate large amounts of information. The whole process, from data storage to drawing conclusions, requires complex computer operations, and so presents a great challenge in the search for new algorithms and methods. Therefore, animal breeding is a multidisciplinary scientific field that relies on achievements from genetics, computer science and statistical mathematics. Additionally, the collection, analysis and dissemination of biological data fall within the scope of bioinformatics. In bioinformatics, mathematical, statistical and computational methods are aimed at solving biological problems using genomic data and related information. Genome sequencing is the first step towards understanding genome organization and gene structure, and in addition, it is necessary to develop techniques for quantifying gene expression through complex computer-based modeling. The implementation of complex analytical methods requires the proper application of mathematical and statistical knowledge to the study of biological systems. Based on the above, it can be said that genomic selection is an example of how the advantages of bioinformatics are used in animal husbandry in order to improve the production system.

Genomic selection implies making selection decisions based on genomically estimated breeding values (gEBVs). gEBVs are calculated using an estimate of SNP effects from a prediction equation, which is derived from animals in a reference (train set) group, and which have certain SNP genotypes and phenotypes for the traits of interest. The accuracy of the gEBV depends on the size of the reference population used to calculate the prediction equation, the heritability of the traits, and the degree of relatedness between the candidate for selection and the reference population [4]. The mostly used methods for genomic prediction include best linear unbiased prediction, and several different models are available, such as TABLUP, RRBLUP, GBLUP, ssGBLUP, and ssSNPBLUP. Ref. [5] pointed out that the TABLUP method provides better results compared to RRBLUP and GBLUP because the TA matrix takes into account one marker that explains to a greater extent the genetic variance for a certain trait, while the matrix is created based on all other data from the pedigree and markers regardless of their influence on a given trait. Ref. [6] studied two dairy cattle populations (German-Austrian-Czech Fleckvieh and German-Austrian Brown Swiss) and concluded that the application of ssGBLUP clearly improves genomic evaluation. Ref. [7] pointed out that ssGBLUP uses all available phenotype, genotype and pedigree information to provide unbiased genomic prediction with higher accuracy than other genomic prediction models. Furthermore, those authors reported that ssSNPBLUP proved to be the most efficient of all variants of the ssGBLUP model, highlighting its advantage in large database applications, such as data processing of up to a million genotyped animals.

The dominant dairy breed in the registered cattle population in Serbia is the Holstein Friesian. The largest part of the population is breeding on the territory of AP Vojvodina, where, according to the data of the Central Breeding Organization for the year 2021, there were registered 67,671 heads [8]. According to the data from this report, the average milk production of Holstein Friesian cows in AP Vojvodina during 305 days of lactation in 2021 was 7405 kg. By comparing this value with the milk yield of recorded cows in other ICAR member countries [9], we see that there are opportunities for improving the traits in the Serbian dairy population, and one of the ways is through the improvement of the methods of the selection of cows. This research aims to evaluate the possibility of introducing the most modern methods for the prediction of breeding values in dairy cattle breeding programs in Serbia. In this regard, the objectives of this research are to: 1) test the implementation of the GBLUP and ssGBLUP methods for calculating breeding values instead of standard uni- and multivariate BLUP models; 2) compare the accuracy of breeding value estimates obtained by the applied methods if only the first lactation is observed; and 3) compare the accuracy of breeding value estimates using the applied methods and when using all lactations, i.e., repeated measurements. Based on the obtained results, recommendations will be defined for changing the existing breeding programs in terms of assessing national breeding values and ranking cattle.

## 2. Materials and Methods

### 2.1. Data Sets and Quality Control

The initial database consisted of 1600 Holstein Friesian cows genotyped with the GeneSeek GGP Bovine 100 K SNP chip, for which data were collected on milk yield traits (MY—milk yield; FY—milk fat yield; FC—milk fat content; PY—milk protein yield; PC—milk protein content). The production results referred to lactation milk production standardized to 305 days using ICAR approved methods for calculating daily yields from AM/PM milkings [10,11] and were provided by the Central Breeding Organization, Department of Animal Husbandry, Faculty of Agriculture Novi Sad.

#### 2.1.1. Preparation of SNP File 

The first processing of SNPs meant turning off the chips for the so-called Mendelian traits, as well as turning off the double SNPs with a smaller GeneTrain parameter. This was followed by the preparation of SNPs for use in the Wombat software [12], which does not accept markers with missing values. For this purpose, some SNPs and animals were eliminated, after which the database consisted of 1566 animals with 62,807 SNPs, each with a call rate of 100%. The next step included SNP quality control in the preGSF90 software [13], whereby one SNP with Mendelian conflicts (MC) and 4091 SNPs with a minor allele frequency of <0.05 were removed. Those animals (*n* = 14) with parent-progeny Mendelian conflicts were also removed. The database after quality control included 1491 animals with 58,715 SNPs.

#### 2.1.2. Preparation of Pedigree Files

Three files were prepared based on origin data for two generations of ancestors. The first file is related to the standard pedigree file for calculating the numerical relationship matrix (***A***) from the pedigree, which is organized through three columns (animal, father, mother). A second pedigree file was created to calculate a genomic relationship matrix (***G***), which is organized through four columns and includes ID numbers for only genotyped animals. The first three columns refer to data on origin (animal, father, mother), and the fourth column indicates that there is a SNP marker for the animal. The third pedigree file was a combination of the previous two files and was used to calculate a hybrid relationship matrix (***H***), which used both pedigree data and SNP data to calculate relationships. It was also organized through four columns with the same meaning as in the second file.

#### 2.1.3. Preparation of Data Files for Measurements of Traits and Factors 

First, two files were prepared for a total of 2565 animals, i.e., cows from the pedigree file A, for which there were measurement results for the investigated production traits. One file related to the cows for which there were data on production during the first lactation, while the second file included repeated measurements, i.e., production results from other lactations as well (I lactation—2565 measurements; II—1448 measurements; III—768 measurements; IV—344; V—116; VI—40). Lactations refer to calvings from 2012 to 2021, which were grouped into three seasons (I—November, December, January, February; II—March, April, September, October; III—May, June, July, August). Then, a third file was prepared, which included the aforementioned data, but only for genotyped animals.

Table 1 provides an overview of the number of animals included in the input files that participated in the trials.

### 2.2. Statistical Analysis of Data

BLUP is often used in animal breeding and is a method that evaluates random effects, which in animal breeding, are most often related to evaluations of breeding values. In all the presented models, the values suggested by Wombat [12] were used for the initial values of the variance components. These are actually variances that were calculated according to well-known formulae. Covariances of *A* and *B* are calculated based on the formula $\hat{cov}(A,B)=r_{A,B}\times \sqrt{s_{A}^{2}}\times \sqrt{s_{B}^{2}}$ where $r_{A,B}$ is the correlation coefficient and $s_{A}^{2}$ and $s_{B}^{2}$ are variances of *A* and *B*. Final values of sample variances and sample covariances were obtained by REML estimation of (co)variance components. Additionally, in all analyses, the fixed part of the model included the influence of the interaction between farm, year and season (*FYS*), and in the model with repeated measurements (*Fix_2_*), the influence of lactation number (L) was additionally included, as follows:*Fix*_*1*_: *y*_*ij*_ = *μ* + *FYS*_*i*_ + *e*_*ij*_
*Fix*_*2*_: *y*_*ijk*_ = *μ* + *FYS*_*i*_ + *L*_*j*_ + *e*_*ijk*_

#### 2.2.1. Univariate Models

##### Prediction of Breeding Values Using Numerical Relationship Matrix (NRM), Single Random Effect Model (Model 1—BLUP)

Ref. [14] first developed BLUP, by which fixed effects and breeding values can be estimated simultaneously. BLUP has found wide use in the genetic evaluation of domestic animals due to its statistical properties. With the constant progress of computers, its application has expanded from simpler models, such as the father model, to more complex models, such as the animal model and its derivatives.

We considered the following equation for a mixed linear model:(1)y=Xb+Za+e
where: y is the vector of observations (features) dimensions n×1, and n is the number of data; b is the vector of unknown fixed effects of dimension p×1, and p is the number of levels of fixed effects; a is the vector of unknown random effects (breeding values) of dimension q×1, and q is the number of levels of random effects; e is the vector of random error effects of dimension n×1; X is a known design matrix of dimension n×p relating the given data to the fixed effects; Z is a known design matrix of dimension n×q relating the given data to the random effects. Matrices X and Z are also called incidence matrices and their elements are 0 and 1.

As in ref. [15], we assumed that the expectations are actually: $E\left(y\right)=Xb$, $E\left(a\right)=E\left(e\right)=0$, and as the error effects are independently distributed with variance $\sigma_{e}^{2}$, $var\left(e\right)=I\sigma_{e}^{2}=R$, $var\left(a\right)=A\sigma_{a}^{2}=F$ i $cov\left(a,e\right)=cov\left(e,a\right)=0$, where A is a matrix of relationship dimension q×q, and I is the unit matrix of dimension n×n. That is, when
$$var\left(y\right)=ZAZ^{\prime }\sigma_{a}^{2}+I\sigma_{e}^{2},\;\;\;\;cov\left(y,a\right)=ZA\sigma_{a}^{2},\;\;\;\;cov\left(y,e\right)=I\sigma_{e}^{2}$$

$Z^{\prime }$ denotes the transposed matrix Z. Mathematically speaking, BLUP provides an estimate for b that we denoted by $\hat{b}$, and an estimate for a, denoted by $\hat{a}$, and which were obtained by solving the equations (MME—Mixed Model Equations) which are, for Equation (1), given as
(2)$$\left[\begin{array}{cc} X^{\prime }R^{-1}X & X^{\prime }R^{-1}Z\\ Z^{\prime }R^{-1}X & Z^{\prime }R^{-1}Z+F^{-1} \end{array}\right]\left[\begin{array}{c} \hat{b}\\ \hat{a} \end{array}\right]=\left[\begin{array}{c} X^{\prime }R^{-1}y\\ Z^{\prime }R^{-1}y \end{array}\right]$$
assuming that the matrices R and F are regular. Note that when $F^{-1}$ tends to the zero matrix, then Equations (2) become a generalized least-squares equations (LSE), and then both b and a are considered as fixed effects [16]. Based on the earlier definition of the matrix R, Equation (2) become
(3)$$\left[\begin{array}{cc} X^{\prime }X & X^{\prime }Z\\ Z^{\prime }X & Z^{\prime }Z+{\alpha_{1}A}^{-1} \end{array}\right]\left[\begin{array}{c} \hat{b}\\ \hat{a} \end{array}\right]=\left[\begin{array}{c} X^{\prime }y\\ Z^{\prime }y \end{array}\right]$$
where $\alpha_{1}=\sigma_{e}^{2}/\sigma_{a}^{2}$. If the matrix in Equations (3) is singular, then a generalized inverse matrix can be used. Also, if the matrix in Equations (3) is written in the form $\left[\begin{array}{cc} U_{11} & U_{12}\\ U_{21} & U_{22} \end{array}\right]$, and with $\left[\begin{array}{cc} U^{11} & U^{12}\\ U^{21} & U^{22} \end{array}\right]$ its generalized inverse matrix, then the accuracy (r) of the breeding values prediction is obtained by the previously published formula [15], $r=\sqrt{1-u_{i}\alpha_{1}}$, where $u_{i}$ is the *i*-th diagonal element of the matrix $U^{22}$.

##### Prediction of Breeding Values Using Numerical Relationship Matrix (NRM), Model with Two Random Effects (Model 2—BLUPp)

The repeatability model was used to analyze data when multiple measurements of the same trait, such as milk yield in consecutive lactations, were recorded in an individual [17].

For models with repetition, the following form was used:(4)y=Xb+Za+Wp+e
where: y is the vector of observations, b is the vector of unknown fixed effects, a is the vector of unknown random effects, p is the vector of unknown environmental random effects of dimension r×1, and e is the vector of error effects. Furthermore, X and Z are the design matrices for the given data regarding fixed and random effects, respectively. The matrix W is a design matrix related to the effects of the external environment (i.e., only those animals for which we had repeated measurements were observed) of dimension n×r.

The vector a contains only the additive random effects of the animal, while the non-additive genetic effects are contained in the vector p. The assumption is that environmental effects and error effects are independently distributed with a mean of zero and variances $\sigma_{p}^{2}$ and $\sigma_{e}^{2}$, respectively. That is when
$$var\left(p\right)=I\sigma_{p}^{2},\;var\left(e\right)=I\sigma_{e}^{2}=R,\;var\left(a\right)=A\sigma_{a}^{2}$$
and
$$var\left(y\right)=var\left(Za+Wp+e\right)=Zvar\left(a\right)Z^{\prime }+Wvar\left(p\right)W^{\prime }+var\left(e\right)\;=ZAZ^{\prime }\sigma_{a}^{2}+WW^{\prime \sigma_{p}^{2}}+I\sigma_{e}^{2}$$

The phenotypic variance $\sigma_{y}^{2}$ is equal to $\sigma_{a}^{2}+\sigma_{p}^{2}+\sigma_{e}^{2}$, while the repeatability coefficient is equal to ${R=(\sigma }_{a}^{2}+\sigma_{p}^{2})/\sigma_{y}^{2}$.

The MME model for Equation (4) is of the form
(5)$$\left[\begin{array}{ccc} X^{\prime }R^{-1}X & X^{\prime }R^{-1}Z & X^{\prime }R^{-1}W\\ Z^{\prime }R^{-1}X & Z^{\prime }R^{-1}Z+\frac{1}{\sigma_{a}^{2}}A^{-1} & Z^{\prime }R^{-1}W\\ W^{\prime }R^{-1}X & W^{\prime }R^{-1}Z & W^{\prime }R^{-1}W+\frac{1}{\sigma_{p}^{2}}I \end{array}\right]\left[\begin{array}{c} \hat{b}\\ \hat{a}\\ \hat{p} \end{array}\right]=\left[\begin{array}{c} X^{\prime }R^{-1}y\\ Z^{\prime }R^{-1}y\\ W^{\prime }R^{-1}y \end{array}\right]$$

After replacing matrix R with matrix $I\sigma_{e}^{2}$ from Equations (5) we get
(6)$$\left[\begin{array}{ccc} X^{\prime }X & X^{\prime }Z & X^{\prime }W\\ Z^{\prime }X & Z^{\prime }Z+\alpha_{1}A^{-1} & Z^{\prime }W\\ W^{\prime }X & W^{\prime }Z & W^{\prime }W+\alpha_{2}I \end{array}\right]\left[\begin{array}{c} \hat{b}\\ \hat{a}\\ \hat{p} \end{array}\right]=\left[\begin{array}{c} X^{\prime }y\\ Z^{\prime }y\\ W^{\prime }y \end{array}\right]$$
where $\alpha_{1}=\sigma_{e}^{2}/\sigma_{a}^{2}$ and $\alpha_{2}=\sigma_{e}^{2}/\sigma_{p}^{2}$. The aim was to evaluate the effects of number of lactations and to predict the breeding values for the animals and the lasting effect of the environment on the observed cows. If the matrix in Equations (6) is written in the form $\left[\begin{array}{ccc} U_{11} & U_{12} & U_{13}\\ U_{21} & U_{22} & U_{23}\\ U_{31} & U_{32} & U_{33} \end{array}\right]$, and with $\left[\begin{array}{ccc} U^{11} & U^{12} & U^{13}\\ U^{21} & U^{22} & U^{23}\\ U^{31} & U^{32} & U^{33} \end{array}\right]$ being its generalized inverse matrix, then the accuracy (r) of the prediction of breeding values is obtained using the formula $r=\sqrt{1-u_{i}\alpha_{1}}$, where $u_{i}$ *i*-th is the diagonal element of the matrix $U^{22}$.

##### Prediction of Breeding Values Using Genomic Relationship Matrix (GRM), Single Random Effect Model (Model 3—GBLUP)

First, we will explain the algorithm for obtaining the genomic relationship matrix [18] assuming the matrix $M_{012}$ is of dimensions q×m, where is q the number of animals, and m is the number of marker columns, i.e., locuses (number of SNPs). If the alleles are A and B and if AA = 0, AB = 1 and BB = 2, then the matrix $M_{012}$ consists of the elements 0, 1 and 2. After its centering, we get the matrix M, that is $M=M_{012}-J$, where J is a matrix of dimension q×m and all elements of which are equal to 1. The elements of matrix M are the numbers −1, 0 and 1. For example, the main diagonal of the matrix $MM^{\prime }$ shows the number of homozygotes for each individual, and outside the diagonal is the number of alleles they share with relatives.

It is necessary to determine the frequencies of allele A and allele B in each of the columns in matrix M. First, in each column in matrix , we determined how many times the numbers −1, 0 and 1 were repeated. Then, $p_{j,-1}$, $p_{j,0}$ and $p_{j,1}$ marked the number of occurrences of the numbers −1, 0 and 1 in the j-th column of matrix M, respectively, j=1,2,…,m. The frequency $p_{j,A}$ of allele A and the frequency $p_{j,B}$ of allele B in the j-th column were obtained using the formula
$$p_{j,A}=\frac{2p_{j,-1}+p_{j,0}}{2q},\;\;\;p_{j,B}=\frac{2p_{j,1}+p_{j,0}}{2q},\;\;\;\;j=1,2,\ldots ,m.$$

Now, matrix $P={\left[p_{ij}\right]}_{q\times m}$ is formed, where its elements are of the form
$$p_{i,j}=2\left(p_{j,B}-0.5\right),i=1,2,\ldots ,q,j=1,2,\ldots ,m$$

Finally, the matrix S is of the form S=M-P. This means that the mean of the elements in each column of the matrix S is exactly 0. One way to obtain the genomic relationship matrix G is to use the following formula [18]
$$G=\frac{SS^{\prime }}{2\sum_{j=1}^{m}p_{j,B}(1-p_{j,B})}$$

By dividing $SS^{\prime }$ by $2\sum_{j=1}^{m}p_{j,B}(1-p_{j,B})$, the matrix G becomes analogous to the relationship matrix A. The genomic inbreeding coefficient for the i-th individual is obtained by subtracting 1 from the i-th element on the main diagonal of matrix G.

As in [19], we wrote the vector of breeding values a as a function of the marker effect, a=Sm. With $var\left(m\right)=I\sigma_{m}^{2}$, we denoted the variance of the marker vector m [20]. Assuming the same variance per locus, variance of a is
(7)$$var\left(a\right)=Svar\left(m\right)S^{\prime }=SS^{\prime }\sigma_{m}^{2}$$

As the genetic variance is equal [21]
(8)$$\sigma_{a}^{2}=2\sum_{j=1}^{m}p_{j,B}(1-p_{j,B})\sigma_{m}^{2}$$
then it follows from Equation (8) that
(9)$$\sigma_{m}^{2}=\frac{\sigma_{a}^{2}}{2\sum_{j=1}^{m}p_{j,B}(1-p_{j,B})}$$
and after including Equation (9) in (7), we have
(10)$$var\left(a\right)=\frac{SS^{\prime }}{2\sum_{j=1}^{m}p_{j,B}(1-p_{j,B})}\sigma_{a}^{2}$$

As calculated by another author [18], the genomic relationship matrix (G) is given by
(11)$$G=\frac{SS^{\prime }}{2\sum_{j=1}^{m}p_{j,B}(1-p_{j,B})}$$
with dimensions q×q, and then, based on Equations (10) and (11)
$${var\left(a\right)=G\sigma }_{a}^{2}$$

The basic model is the same as in Equation (1), where $var\left(e\right)=I\sigma_{e}^{2}=R$. An appropriate MME model is
$$\left[\begin{array}{cc} X^{\prime }X & X^{\prime }Z\\ Z^{\prime }X & Z^{\prime }Z+{\alpha_{1}G}^{-1} \end{array}\right]\left[\begin{array}{c} \hat{b}\\ \hat{a} \end{array}\right]=\left[\begin{array}{c} X^{\prime }y\\ Z^{\prime }y \end{array}\right]$$
where is $\alpha_{1}=\sigma_{e}^{2}/\sigma_{a}^{2}$.

If the matrix G is singular, then it is replaced by $G_{r}$ [22], where
$$G_{r}=\lambda \left(\beta \left(G+\varepsilon I\right)+\alpha J\right)+\left(1-\lambda \right)A$$
with 0≤λ≤1, we denoted the ratio of the total genetic variance and the marker effect, A, as the relationship matrix, α, and β as the leveling factors proposed in [23] and [24], J is a square matrix with all elements equal to 1, I is a unit matrix and 0<ε≪1 is a small constant. Thus, GBLUP is BLUP, where A is replaced by the genomic matrix G, i.e., $G_{r}$.

##### Prediction of Breeding Values Using Genomic Relationship Matrix (GRM), All Animals Were Genotyped, Model with Two Random Effects (Model 4—GBLUPp)

The basic model was then Equation (4) with $var\left(e\right)=I\sigma_{e}^{2}=R$, $var\left(a\right)=G\sigma_{a}^{2}$ and $var\left(p\right)=I\sigma_{p}^{2}$. The MME model is
(12)$$\left[\begin{array}{ccc} X^{\prime }X & X^{\prime }Z & X^{\prime }W\\ Z^{\prime }X & Z^{\prime }Z+\alpha_{1}G^{-1} & Z^{\prime }W\\ W^{\prime }X & W^{\prime }Z & W^{\prime }W+\alpha_{2}I \end{array}\right]\left[\begin{array}{c} \hat{b}\\ \hat{a}\\ \hat{p} \end{array}\right]=\left[\begin{array}{c} X^{\prime }y\\ Z^{\prime }y\\ W^{\prime }y \end{array}\right]$$
where $\alpha_{1}=\sigma_{e}^{2}/\sigma_{a}^{2}$ and $\alpha_{2}=\sigma_{e}^{2}/\sigma_{p}^{2}$. If the matrix in Equations (12) is singular, its generalized inverse matrix is used when obtaining $\hat{b}$, $\hat{a}$ and $\hat{p}$.

##### Prediction of Breeding Values Using Genomic Relationship Matrix (GRM), and Including Animals That Were Not Genotyped, Single Random Effect Model (Model 5—ssGBLUP)

The idea for ssGBLUP arose from the fact that a smaller part of the animals from the observed population was genotyped. A relationship matrix of the form A is observed
$$A=\left[\begin{array}{cc} A_{11} & A_{12}\\ A_{21} & A_{22} \end{array}\right]$$
where index number 1 refers to animals that have not been genotyped, and index number 2 to animals that have been genotyped. Now, the matrix $M_{g012}$ has dimensions $q_{g}\times m$, where $q_{g}$ is the number of genotyped animals and m is the number of marker columns (i.e., the number of SNPs). After centering the matrix $M_{g012}$, we get the matrix $M_{g}$, whose elements are −1, 0 and 1. After determining the allele frequency (as in the previous section), the matrix S will be of the form $S=M_{g}-P$. The genomic relationship matrix ($G_{g}$) is given by
$$G_{g}=\frac{SS^{\prime }}{2\sum_{j=1}^{m_{g}}p_{j,B}(1-p_{j,B})}$$
where $p_{j,B}$ is the frequency of allele B in the j-th column of the SNP. If the matrix $G_{g}$ is singular, then it is replaced by $G_{gr}$ where
$$G_{gr}=\lambda \left(\beta \left(G_{g}+\varepsilon I\right)+\alpha J\right)+\left(1-\lambda \right)A_{22}$$

Clearly, $A_{22}$ is a matrix of dimension $q_{g}\times q_{g}$ and is the relationship matrix only for genotyped animals. Matrices J and I were defined in the previous section. The matrix $H^{-1}$ is obtained as follows [25]
$$H^{-1}=A^{-1}+\left[\begin{array}{cc} 0 & 0\\ 0 & G_{gr}^{-1}-A_{22}^{-1} \end{array}\right]$$

The MME model for Equation (1) is
(13)$$\left[\begin{array}{cc} X^{\prime }X & X^{\prime }Z\\ Z^{\prime }X & Z^{\prime }Z+{\alpha_{1}H}^{-1} \end{array}\right]\left[\begin{array}{c} \hat{b}\\ \hat{a} \end{array}\right]=\left[\begin{array}{c} X^{\prime }y\\ Z^{\prime }y \end{array}\right]$$
where are $\alpha_{1}=\sigma_{e}^{2}/\sigma_{a}^{2}$, $var\left(e\right)=I\sigma_{e}^{2}=R$, $var\left(a\right)=H\sigma_{a}^{2}$. Thus, ssGBLUP is BLUP, where A is replaced by the genomic matrix H. Note that if all animals are genotyped, $H^{-1}=G_{gr}^{-1}$.

##### Prediction of Breeding Values Using Genetic Markers (GRM), Existing Animals Not Genotyped, Model with Two Random Effects (Model 6—ssGBLUPp)

The basic model is Equation (4) with $var\left(e\right)=I\sigma_{e}^{2}=R$, $var\left(a\right)=H\sigma_{a}^{2}$ and $var\left(p\right)=I\sigma_{p}^{2}$. The MME model is
(14)$$\left[\begin{array}{ccc} X^{\prime }X & X^{\prime }Z & X^{\prime }W\\ Z^{\prime }X & Z^{\prime }Z+\alpha_{1}H^{-1} & Z^{\prime }W\\ W^{\prime }X & W^{\prime }Z & W^{\prime }W+\alpha_{2}I \end{array}\right]\left[\begin{array}{c} \hat{b}\\ \hat{a}\\ \hat{p} \end{array}\right]=\left[\begin{array}{c} X^{\prime }y\\ Z^{\prime }y\\ W^{\prime }y \end{array}\right]$$
where $\alpha_{1}=\sigma_{e}^{2}/\sigma_{a}^{2}$ and $\alpha_{2}=\sigma_{e}^{2}/\sigma_{p}^{2}$. In the case of a singular matrix for this model, its generalized inverse matrix was used.

#### 2.2.2. Multivariate Models

In this study, we had some dairy cow data that encompassed only the first lactation, but other data encompassing multiple lactations. Therefore, we defined MBLUP and MBLUPp methods that referred only to the first lactation and to all lactations, respectively, but each method simultaneously analyzed all five traits.

##### Prediction of Breeding Values by Multivariate Analysis Using Relationship Matrix (NRM), Single Random Effect Model (Model 7—MBLUP)

The model for multivariate analysis is reduced to the grouping of multiple univariate models for each of the traits. If the number of traits is *np*, then the univariate model for each of the *np* traits is $y_{i}=X_{i}b_{i}+Z_{i}a_{i}+e_{i}$ for i=1,2,…,np. Here, $y_{i}$ is the observation vector for the i-th trait, $b_{i}$ is the vector of fixed effects for a trait i; $a_{i}$ is the vector of random effects of animals for a trait i, $e_{i}$ is the vector of random effects of error for i-th trait. $X_{i}$ and $Z_{i}$ are known matrices of the design that relate to the given data to fixed effects and random effects, respectively, and also for trait ii. In matrix form, a multivariate model can be written in the form
(15)$$\left[\begin{array}{c} y_{1}\\ y_{2}\\ \vdots \\ y_{np} \end{array}\right]=\left[\begin{array}{cccc} X_{1} & 0 & \cdots & 0\\ 0 & X_{2} & \cdots & 0\\ \vdots & \vdots & \ddots & \vdots \\ 0 & 0 & \cdots & X_{np} \end{array}\right]\left[\begin{array}{c} b_{1}\\ b_{2}\\ \vdots \\ b_{np} \end{array}\right]+\left[\begin{array}{cccc} Z_{1} & 0 & \cdots & 0\\ 0 & Z_{2} & \cdots & 0\\ \vdots & \vdots & \ddots & \vdots \\ 0 & 0 & \cdots & Z_{np} \end{array}\right]\left[\begin{array}{c} a_{1}\\ a_{2}\\ \vdots \\ a_{np} \end{array}\right]+\left[\begin{array}{c} e_{1}\\ e_{2}\\ \vdots \\ e_{np} \end{array}\right]$$

We assumed the version reported previously [15].
$$var\left[\begin{array}{c} a_{1}\\ a_{2}\\ \vdots \\ a_{np}\\ e_{1}\\ e_{2}\\ \vdots \\ e_{np} \end{array}\right]=\left[\begin{array}{cccccccc} g_{11}A & g_{12}A & \cdots & g_{1,np}A & 0 & 0 & \cdots & 0\\ g_{21}A & g_{22}A & \cdots & g_{2,np}A & 0 & 0 & \cdots & 0\\ \vdots & \vdots & \ddots & \vdots & \vdots & \vdots & \ddots & \vdots \\ g_{np,1}A & g_{np,2}A & \cdots & g_{np,np}A & 0 & 0 & \cdots & 0\\ 0 & 0 & \cdots & 0 & r_{11}I & r_{12}I & \cdots & r_{1,np}I\\ 0 & 0 & \cdots & 0 & r_{21}I & r_{22}I & \cdots & r_{2,np}I\\ \vdots & \vdots & \cdots & \vdots & \vdots & \vdots & \ddots & \vdots \\ 0 & 0 & \cdots & 0 & r_{np,1}I & r_{np,2}I & \cdots & r_{np,np}I \end{array}\right]$$
where: $G={\left[g_{ij}\right]}_{np\times np}$ is the additive genetic matrix of variances and covariances of animal effects where, for example, $g_{11}$ is the additive genetic variance of the first trait, $g_{12}=g_{21}$ is the additive genetic covariance between the first and second traits, etc.; A is a relationship matrix;$R={\left[r_{ij}\right]}_{np\times np}$ is a matrix of variances and covariances of error effects.

The MME model for Equations (15) is
(16)$$\left[\begin{array}{cc} X^{\prime }R^{-1}X & X^{\prime }R^{-1}Z\\ Z^{\prime }R^{-1}X & Z^{\prime }R^{-1}Z+A^{-1}{⨂G}^{-1} \end{array}\right]\left[\begin{array}{c} \hat{b}\\ \hat{a} \end{array}\right]=\left[\begin{array}{c} X^{\prime }R^{-1}y\\ Z^{\prime }R^{-1}y \end{array}\right]$$
where
(17)$$X=\left[\begin{array}{cccc} X_{1} & 0 & \cdots & 0\\ 0 & X_{2} & \cdots & 0\\ \vdots & \vdots & \ddots & \vdots \\ 0 & 0 & \cdots & X_{np} \end{array}\right],Z=\left[\begin{array}{cccc} Z_{1} & 0 & \cdots & 0\\ 0 & Z_{2} & \cdots & 0\\ \vdots & \vdots & \ddots & \vdots \\ 0 & 0 & \cdots & Z_{np} \end{array}\right],\hat{b}=\left[\begin{array}{c} {\hat{b}}_{1}\\ {\hat{b}}_{2}\\ \vdots \\ {\hat{b}}_{np} \end{array}\right],\hat{a}=\left[\begin{array}{c} {\hat{a}}_{1}\\ {\hat{a}}_{2}\\ \vdots \\ {\hat{a}}_{np} \end{array}\right],y=\left[\begin{array}{c} y_{1}\\ y_{2}\\ \vdots \\ y_{np} \end{array}\right]$$

If $G^{-1}={\left[g^{ij}\right]}_{np\times np}$, then
$$A^{-1}{⨂G}^{-1}={\left[{A^{-1}g}^{ij}\right]}_{np\times np}={\left[\begin{array}{cccc} {A^{-1}g}^{11} & {A^{-1}g}^{12} & \cdots & {A^{-1}g}^{1,np}\\ {A^{-1}g}^{21} & {A^{-1}g}^{22} & \cdots & {A^{-1}g}^{2,np}\\ \vdots & \vdots & \ddots & \vdots \\ {A^{-1}g}^{np,1} & {A^{-1}g}^{np,2} & \cdots & {A^{-1}g}^{np,np} \end{array}\right]}_{np\times np}$$

##### Prediction of Breeding Values by Multivariate Analysis Using Relationship Matrix (NRM), Model with Two Random Effects (Model 8—MBLUPp)

In matrix form, a multivariate model (*np* traits) with environmental effects was written as
(18)$$\left[\begin{array}{c} y_{1}\\ y_{2}\\ \vdots \\ y_{np} \end{array}\right]=\left[\begin{array}{cccc} X_{1} & 0 & \cdots & 0\\ 0 & X_{2} & \cdots & 0\\ \vdots & \vdots & \ddots & \vdots \\ 0 & 0 & \cdots & X_{np} \end{array}\right]\left[\begin{array}{c} b_{1}\\ b_{2}\\ \vdots \\ b_{np} \end{array}\right]+\left[\begin{array}{cccc} Z_{1} & 0 & \cdots & 0\\ 0 & Z_{2} & \cdots & 0\\ \vdots & \vdots & \ddots & \vdots \\ 0 & 0 & \cdots & Z_{np} \end{array}\right]\left[\begin{array}{c} a_{1}\\ a_{2}\\ \vdots \\ a_{np} \end{array}\right]+\left[\begin{array}{cccc} W_{1} & 0 & \cdots & 0\\ 0 & W_{2} & \cdots & 0\\ \vdots & \vdots & \ddots & \vdots \\ 0 & 0 & \cdots & W_{np} \end{array}\right]\left[\begin{array}{c} {pe}_{1}\\ {pe}_{2}\\ \vdots \\ {pe}_{np} \end{array}\right]+\left[\begin{array}{c} e_{1}\\ e_{2}\\ \vdots \\ e_{np} \end{array}\right]$$
where $W_{i}$ is the design matrix related to the effects of the external environment for the *i*-th trait. If $C={\left[c_{ij}\right]}_{np\times np}$ is the matrix of variances and covariances of external environmental effects, the MME model for Equations (18) is
(19)$$\left[\begin{array}{ccc} X^{\prime }R^{-1}X & X^{\prime }R^{-1}Z & X^{\prime }R^{-1}W\\ Z^{\prime }R^{-1}X & Z^{\prime }R^{-1}Z+A^{-1}{⨂G}^{-1} & Z^{\prime }R^{-1}W\\ W^{\prime }R^{-1}X & W^{\prime }R^{-1}Z & W^{\prime }R^{-1}W+I{⨂C}^{-1} \end{array}\right]\left[\begin{array}{c} \hat{b}\\ \hat{a}\\ \hat{p} \end{array}\right]=\left[\begin{array}{c} X^{\prime }R^{-1}y\\ Z^{\prime }R^{-1}y\\ W^{\prime }R^{-1}y \end{array}\right]$$
where, along with (17),
$$W=\left[\begin{array}{cccc} W_{1} & 0 & \cdots & 0\\ 0 & W_{2} & \cdots & 0\\ \vdots & \vdots & \ddots & \vdots \\ 0 & 0 & \cdots & W_{np} \end{array}\right],\hat{p}=\left[\begin{array}{c} {\hat{p}}_{1}\\ {\hat{p}}_{2}\\ \vdots \\ {\hat{p}}_{np} \end{array}\right]$$

If $C^{-1}={\left[c^{ij}\right]}_{np\times np}$, then $I{⨂C}^{-1}={\left[{Ic}^{ij}\right]}_{np\times np}$.

### 2.3. Two-Tailed t-Test

A two-tailed *t*-test [26] was performed to check statistically significant differences in the average estimation of accuracy and breeding values between two selected models. Suppose we have two independent normal populations with unknown means $\mu_{1}$ and $\mu_{2}$, and unknown but equal variances. Let a random sample from the first population have $n_{1}$ observations and a random sample from the second population have $n_{2}$ observations. Let ${\overset{-}{X}}_{1}$, ${\overset{-}{X}}_{2}$, $S_{1}^{2}$ and $S_{2}^{2}$ be the sample means and sample variances, respectively. For
$$S_{p}^{2}=\frac{n_{1}-1}{n_{1}+n_{2}-2}S_{1}^{2}+\frac{n_{2}-1}{n_{1}+n_{2}-2}S_{2}^{2}$$
the quantity
$$t=\frac{{\overset{-}{X}}_{1}-{\overset{-}{X}}_{1}-(\mu_{1}-\mu_{2})}{S_{p}\sqrt{\frac{1}{n_{1}}+\frac{1}{n_{2}}}}$$
has a *t* distribution with $n_{1}+n_{2}-2$ degrees of freedom. We wish to test
$$H_{0}:\mu_{1}-\mu_{2}=0\;H_{1}:\mu_{1}-\mu_{2}\neq 0$$
rejecting the $H_{0}$ if $t>t_{\alpha ,n_{1}+n_{2}-2}$ or $t<-t_{\alpha ,n_{1}+n_{2}-2}$. Here, $t_{\alpha ,n_{1}+n_{2}-2}$ is the value in the *t*-statistic table for level of significance α. In this manuscript, these two populations were estimates of the accuracy of breeding values or estimates of the breeding values for two different mathematical models that have been considered in the case of all traits. The null hypothesis were that there was no statistically significant difference between the average estimate of the accuracy of breeding values or the average estimate of the breeding values regarding the two performed mathematical models.

### 2.4. Spearman’s Rank and Pearson’s Correlation Coefficients

Spearman’s rank correlation [27] measures the strength and direction of association between two ranked variables (breeding values). It basically provides the measure of monotonicity of the relation between two variables, i.e., how well the relationship between two variables could be represented using a monotonic function. The formula for Spearman’s rank coefficient is:$$\rho =1-\frac{6\sum_{i=1}^{n}d_{i}^{2}}{n(n^{2}-1)}$$
where ρ is Spearman’s rank correlation coefficient, $d_{i}$ is the difference between two ranks of each observation, n is the number of observations. The Spearman’s rank correlation can take a value from +1 to −1. A value of +1 means a perfect association of rank. A value of 0 means that there is no association between ranks. Finally, a value of −1 means a perfect negative association of rank.

Pearson’s correlation coefficient [27] measures the strength between the different variables and their relationships. Therefore, whenever any statistical test is conducted between the two variables (breeding values), it is always a good idea for the person analyzing to calculate the value of the correlation coefficient to know how strong the relationship between the two variables is. Let $X_{11},X_{12},\ldots ,X_{1n}$ be the samples of the first variable and $X_{21},X_{22},\ldots ,X_{2n}$ of the second variable. Pearson’s correlation coefficient (R) is
$$R=\frac{n\sum_{i=1}^{n}X_{1i}X_{2i}-\sum_{i=1}^{n}X_{1i}\sum_{i=1}^{n}X_{2i}}{\sqrt{n\sum_{i=1}^{n}X_{1i}^{2}-{\left(\sum_{i=1}^{n}X_{1i}\right)}^{2}}\sqrt{n\sum_{i=1}^{n}X_{2i}^{2}-{\left(\sum_{i=1}^{n}X_{2i}\right)}^{2}}}$$

Pearson’s correlation coefficient returns a value between −1 and 1. The interpretation of the correlation coefficient is: if the correlation coefficient is −1, it indicates a strong negative relationship, implying a perfect negative relationship between the variables; if the correlation coefficient is 0, it indicates no relationship; if the correlation coefficient is +1, it indicates a strong positive relationship, implying a perfect positive relationship between the variables.

## 3. Software Tools for Data Preparation and Processing

Within the BioITGenoSelect project, several software solutions were developed that were used in the preparation of data for this work. Some of the functionality is part of a central application that manages animal data (e.g., origin, lactation), and several programs were written to run independently and can be run from the command line on given input files. 

Genotype data were obtained from the Neogen laboratory in a format standard for Illumina GenomeStudio Software (https://www.illumina.com/techniques/microarrays/array-data-analysis-experimental-design/genomestudio.html, accessed on 12 December 2022). The exports important for the project are in the so-called Final Report format, in which pairs of animals and SNP reads are given in several forms. For the purposes of the work, it was necessary to convert the data into 0125 input form, in which the number of alleles marked with B was counted, while the number 5 represented an unknown value. In the output format, each animal has an associated sequence of numbers representing individual SNP values. Additionally, applications were written that allowed the minimum amount of SNP columns and animals to be removed that would achieve 100% of known values (i.e., 5 does not appear in the output). Since the work used tools from the BLUPF90 family [13] and Wombat [12], which have different input formats for this data, applications were also written that performed translations where necessary.

Data on origin and lactations were generated from the web application BioITGenoSelect (https://bioitgenoselect.polj.uns.ac.rs/genoselect, accessed on 12 December 2022) for the requested animals, based on imported data from the databases curated by the Faculty of Agriculture of the Central Breeding Organization in the livestock sector in Vojvodina province, Serbia. 

### 3.1. PreGSF90

PreGSF90 is an interface program for the genomic module to process the genomic information for the BLUPF90 family of programs. Within the BLUPF90 program package [13], there are a number of applications that can be used in different steps of data processing. The package as a whole is designed for various calculations using mixed models in animal breeding. For the purposes of this project, some of the program’s data quality control applications were used, primarily PreGSF90, in combination with RENUMF90, which was required for some of the inputs.

### 3.2. Wombat

Wombat [12] is a software for analyzing data related to quantitative genetics and animal breeding, and uses linear mixed models with a focus on REML estimation of (co)variance components. It was first presented at the eighth WCGALP 2006, and in the last few years, it has been significantly improved and expanded, primarily with multivariate analyses and analyses for solving mixed model equations using genomic data and ssGBLUP methods. In Serbia, this program is currently used for estimations of breeding values and ranking of cattle for the purposes of implementing national breeding programs, using univariate analysis with the random effect of the animal. In this study, existing analyses were extended to the use of multivariate analysis (MBLUP), as well as the use of models for genomic prediction (GBLUP and ssGBLUP), with the aim of defining and recommending the model that best fits the currently available data.

## 4. Results

BLUP(p), ssGBLUP(p) and MBLUP(p) models were first compared because they provided breeding values for all 6041 animals. After that, BLUP(p), GBLUP(p) and ssGBLUP(p) methods were compared, but only for 1491 genotyped animals because GBLUP(p) methods gave breeding values only for those. As the Wombat software does not support multivariate analysis when using SNP markers, it was applied only in cases where the NRM was used, and not when the GRM was used.

### 4.1. Comparison of BLUP(p), ssGBLUP(p) and MBLUP(p) Models

When solving Equations (3)—BLUP model, (6)—BLUPp model, (13)—ssGBLUP model, (14)—ssGBLUPp model, (16)—MBLUP model, and (19)—MBLUPp model, 2565 animals were used, for which there were data for all five observed traits: milk yield (MY), milk fat yield (FY), milk fat content (FC), milk protein yield (PY) and milk protein content (PC) The pedigree file consisted of 6041 animals, so breeding value estimates were obtained for all 6041 animals. Table 2 provides the accuracies of the evaluations of the obtained breeding values for each observed trait, which were created by solving Equations (3), (6), (13), (14), (16) and (19).

For each observed trait, there was a statistically significant difference in the average estimate of accuracy depending on the method used. The average accuracy was higher using the BLUPp model compared to BLUP, except in the case of the trait PC. The same was true for the ssGBLUPp and MBLUPp models versus ssGBLUP and MBLUP models, respectively. The average increase in accuracy of the BLUPp model compared to the BLUP model for the first four traits was 15.12%, the accuracy increase of ssGBLUPp compared to ssGBLUP was 14.78%, while for the MBLUPp model compared to MBLUP, it was 15.11%. For each trait, the MBLUP model gave higher accuracy than the BLUP model. The same trend was clear for the MBLUPp model in relation to BLUPp (MBLUPp was more accurate), except for the trait FC.

Table 3 shows the mean values of the estimated breeding values obtained by solving the observed systems, as well as the *p*-values of the applied two-tailed *t*-test (the mean values obtained by the BLUP and BLUPp models, ssGBLUP and ssGBLUPp models, and MBLUP and MBLUPp models were compared).

A two-tailed *t*-test showed that there were no statistically significant differences in the average estimates of breeding values when comparing BLUP and BLUPp, except for the trait PC. When the ssGBLUP and ssGBLUPp models were compared, there were statistically significant differences in the average estimates of breeding values for all traits except for the trait FC. When comparing MBLUP and MBLUPp models, there were statistically significant differences in the average estimates of breeding values for all the traits studied.

### 4.2. Estimates of Breeding Values and Comparison of Accuracy of Estimates Using NRM for BLUP(p) Models and GRM for GBLUP(p) and ssGBLUP(p) Models for 1491 Genotyped Animals

Breeding value estimates for 1491 genotyped animals were extracted from data generated by the BLUP(pe) and ssGBLUP(p) models that had provided breeding values for 6041 animals, to compare the accuracy of breeding values estimated by the GBLUP(p) models; these models used data and calculated breeding values only for genotyped animals. Descriptive statistics for these methods are shown in Table 4.

When we consider only the first lactations, the GBLUP model proved to be more accurate than the BLUP model, except for the PC trait. The most accurate method was ssGBLUPp, with a maximum standard error of 0.0406. The average percent increase in milk fat yield when estimated using the ssBLUP model compared to BLUP was 5.01%. If we look at all lactations, for all five traits, the highest estimate of accuracy was obtained by the ssGBLUPp model. With the BLUPp model, the accuracy ranged from 0.540 to 0.753, with GBLUPp from 0.529 to 0.736 and with ssGBLUPp from 0.582 to 0.766. The average percentage increase in accuracy of the ssGBLUPp method was 5.91% over the BLUPpe method. With the ssGBLUPp model, the standard error was not greater than 0.0559.

Table 5 shows the mean estimated breeding values calculated using the BLUP(p), GBLUP(p) and ssGBLUP(p) models.

### 4.3. Correlations between BVs

The correlations between BVs for the BLUP, GBLUP and ssGBLUP models are shown in Table 6. These correlations were performed on 1491 cows because GBLUP and GBLUPp can only be used on genotyped cows.

For all five traits, the highest correlations were between GBLUP and ssGBLUP. The highest Spearman’s rank correlation coefficient is 0.956 (PC trait, GBLUP—ssGBLUP) and the lowest 0.785 (MY trait, BLUP—GBLUP). The highest Pearson’s coefficient is 0.962 (PC trait, GBLUP—ssGBLUP) and the lowest 0.789 (MY trait, BLUP—GBLUP). In Figure 1, for the MY trait, the scatter plots between BVs predicted in BLUP, GBLUP and ssGBLUP are shown.

The correlations between BVs for the BLUPp, GBLUPp and ssGBLUPp models are also shown in Table 7 for 1491 genotyped cows.

In this case, for all traits, the highest correlations were between GBLUPp and ssGBLUPp. The highest Spearman’s rank correlation coefficient is 0.959 (FC trait, GBLUPp—ssGBLUPp) and the lowest 0.702 (PY trait, BLUPp—GBLUPp). The highest Pearson’s coefficient is 0.962 (FC trait, GBLUPp—ssGBLUPp) and the lowest 0.732 (PY trait, BLUPp—GBLUPp). In Figure 2, for the PY trait, the scatter plots between BVs predicted in BLUPp, GBLUPp and ssGBLUPp are shown.

## 5. Discussion

The estimation of breeding values and animal ranking are important parts of modern breeding programs. The accuracy of an estimated breeding value is defined as the correlation between the true and estimated breeding values [28]. This precision is highly dependent on the amount of information available on any single animal, including data on its relatives. Three main sources of information are: the performance data, measured on animals; data on correlations between different traits; and pedigree data, which nowadays are often accompanied by the results of molecular genetic analysis [29]. This enables the implementation of genomic selection that results in great genetic progress and lowers the costs of progeny testing, presenting breeders with the opportunity to select animals that have inherited desirable gene combinations [30,31]. Genomic selection includes the genotyping of animals irrespective of their age and sex, and this allows breeding values to be calculated, even for a newborn male or female animal. This selection approach is recognized by the relevant international organizations, and its advantages in animal breeding are a shorter generation interval, increased selection effect, more accessible data validation and inbreeding prevention. The application of genomic selection in dairy cattle breeding started during the last years of the first decade of the 21st century [32,33], enabled by the analysis of SNP markers that are utilized over the whole genome, and by being relatively inexpensive. 

Breeding value estimation and the ranking of dairy cows in Serbia is currently based on the application of the univariate BLUP animal model, where the greatest emphasis is placed on the milk yield characteristics, primarily on the milk yield in the first lactation [34]. Milk yield traits are of primary economic importance in breeding programs because they directly affect the profitability of cattle production [35]. However, the selection of breeding animals should be based on the combination of traits included in a selection index by which the animals are then ranked. Multivariate analysis is the optimal methodology for the analysis of several traits because it simultaneously evaluates the breeding values of animals and calculates phenotypic and genetic correlations between these traits [15]. In addition, one of the major advantages of multivariate over univariate analysis is that it increases the accuracy of the estimate [15]. While observing the data presented in Table 2, it can be seen that using the multivariate analyses (MBLUP/MBLUPp), higher accuracies of breeding values were obtained, on average, for all traits compared to the univariate analyses (BLUP/BLUPp). Additionally, from Table 2, increases in the accuracies of the estimated breeding values were also achieved by the use of the repeated measurements (milk yield results from repeated lactations), which was confirmed for the majority of traits (except for the protein content), regardless of whether it was a standard univariate or multivariate analysis with the use of a numerical relationship matrix or analysis combining numerical and genomic relationship matrices (ssGBLUP).

It is well known that increasing the amount of information on a single animal moves an estimated breeding value closer to the true breeding value. Table 4 shows that, for all traits included, ssGBLUP(p) produced more accurate estimated breeding values than did BLUP(p) and GBLUP(p), while GBLUP had better accuracy than BLUP. However, data in Table 2 show that the application of the ssGBLUP model only partially improved the accuracy. More precisely, the inclusion of SNP marker data resulted in increased accuracy over the univariate BLUP(p) models, but not over the multivariate BLUP(p) models. It can be concluded that models that include genomic data produce more accurate estimated breeding values for genotyped animals, while the same is not necessarily true at the level of the whole population; we speculate that in our case, this was probably a result of the small number of genotyped animals within the population. Ref. [36] compared single-step blending models and GBLUP models with and without adjustment of the genomic relationship matrix for the genomic prediction of 16 traits in a Nordic Holstein population. The accuracy for traits of milk, fat and protein yields were 0.657, 0.675 and 0.655, respectively, and these results were obtained using the GBLUP model. In the current research, the accuracies of milk, fat and protein yields were 0.523, 0.521 and 0.545, respectively, with the GBLUP model. Ref. [37] compared the accuracies of genomic prediction for milk, fat and protein yields from Philippine dairy buffaloes using BLUP, GBLUP and ssGBLUP. In their research, the ssGBLUP model was 13% more accurate than the GBLUP model for milk yield, while in our research, the difference in milk yield estimates between those same models was 4.2%. For fat yield, [37] found an increase of 4%, while ours was 2.5%, and for protein yield, the increase was 3% in [37], while it was 2.6% in our current study. Similar comparisons were also made in other species. The accuracy of BLUP and ssGBLUP for predicting breeding value in Huacaya Alpacas were compared in [38]. Their model with genomic data was, on average, 3.51% more accurate. In our current study, the ssGBLUP model was 4.90% more accurate than the BLUP model. In [39], authors compared the accuracy of BLUP and GBLUP predictions of immune traits in Beijing oil chickens. Average accuracies were 0.316 and 0.274, using BLUP and GBLUP models, respectively. In our current research, these accuracies were 0.573 and 0.579, using BLUP and GBLUP models, respectively. According to [40] accuracies in genomic selection depend on the number, distribution, and contributions of genotypes and phenotypes to the genomic evaluation and [41] conclude that the design and size of the reference population plays a major role in achieving accuracy for the breeding schemes.

A comparison of predicted breeding values calculated using different analyses (BLUP, GBLUP and ssGBLUP) was also performed using Spearman’s and Pearson’s correlation coefficients, which is in agreement with [42]. In this research, high positive correlations were also obtained in relation to the applied analyses, and the highest were between GBLUP and ssGBLUP, regardless of the investigated trait. Some authors compared genomic prediction using GBLUP and ssGBLUP methods in Iranian Holstein cattle by calculating reliability [43]. In this research of ssGBLUP, the average confidence of genomic predictions for the three traits (milk, fat, and protein yield) was 0.39, which is 0.98% points higher than the average confidence from the GBLUP method.

## 6. Conclusions

The integration of bioinformatics in animal breeding is a new approach in Serbia, offering numerous advantages regarding the accuracy and speed of achieving previously set goals. The development of databases that incorporate pedigree, performance and genotype data results in valuable resources for detecting new candidate genes, linked with traits of economic interest, while also increasing the accuracy of estimated breeding values of animals that are candidates for selection. 

Greater accuracy when using repeated measures over multiple lactations was confirmed for almost all traits. Furthermore, greater accuracy was obtained with the combination of NRM and GRM, i.e., using the ssGBLUP model compared with either NRM or using GRM in BLUP and GBLUP models, but only when focusing just on genotyped animals. At the level of the examined population that combined both genotyped and non-genotyped animals, the highest accuracy was achieved using multivariate analysis with repeated measures. 

In conclusion, this study shows that the current breeding programs in Serbia should be changed to use multivariate analysis with repeated measurements for estimating breeding values and for ranking animals that are candidates for selection, until the optimal size of reference population data, which must include genotyping data on both bulls and cows, is reached. This would have economic significance at the national level because greater accuracy in the prediction of breeding value expands the possibilities for increasing the intensity of selection and leads to an increase in the rate of genetic improvement of economically useful traits in cows.

## Figures and Tables

**Figure 1 animals-13-00597-f001:**
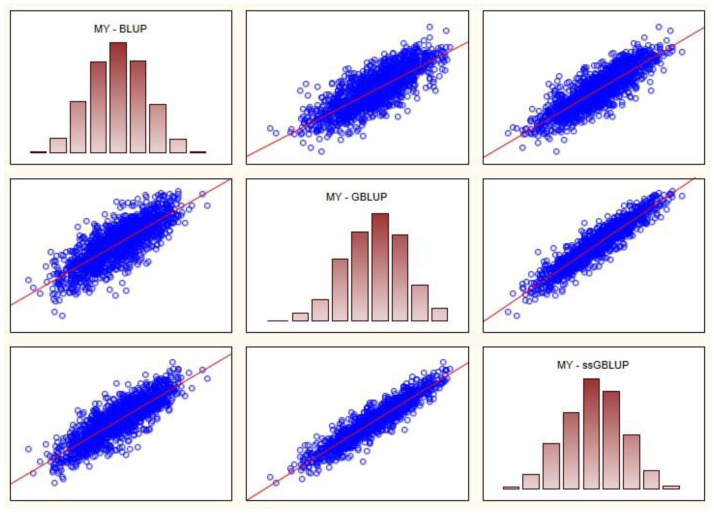
Spearman rank (above diagonal) and Pearson correlations (below diagonal) between predicted breeding values for MY trait.

**Figure 2 animals-13-00597-f002:**
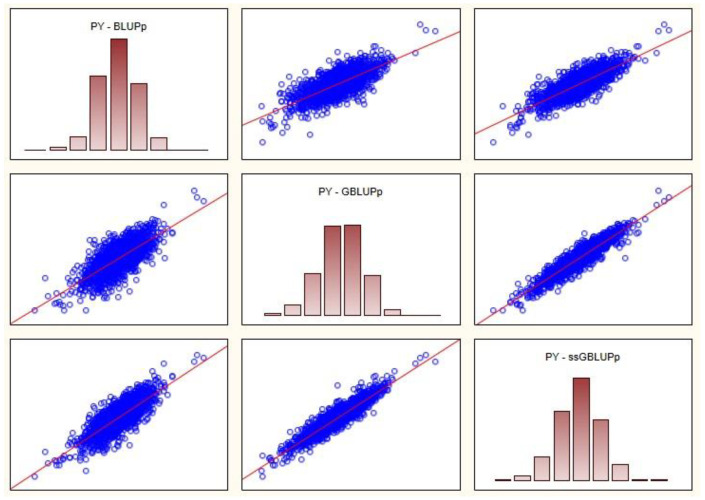
Spearman rank (above diagonal) and Pearson correlations (below diagonal) between predicted breeding values for PY trait.

**Table 1 animals-13-00597-t001:** Number of animals included in the input files.

Number of Animals	Pedigree File		
	*A*	*G*	*H*
Total	6041	1491	6041
Complete pedigree	4087	1491	4087
Measurement results	2565	1491	2565
Genotyped	-	1491	1491

***A***—pedigree file for the NRM matrix of relationships; ***G*** and ***H***—pedigree files for the GRM matrix of relationships.

**Table 2 animals-13-00597-t002:** Descriptive statistics and *p*-values obtained using a two-tailed *t*-test for the accuracy of prediction of breeding values calculated using the BLUP(p), ssGBLUP(p) and MBLUP(p) models.

Trait	BLUP				BLUPp				*p*-Value	An Increase/ Reduction (%)
	$\overset{-}{X}$	SD	Min	Max	$\overset{-}{X}$	SD	Min	Max		
MY	0.334	0.1724	0.035	0.822	0.385	0.1947	0.041	0.877	$9.79\times 10^{-52}$	+15.27
FY	0.349	0.1783	0.037	0.837	0.404	0.2024	0.044	0.89	$1.29\times 10^{-56}$	+15.76
FC	0.429	0.2090	0.051	0.897	0.530	0.2486	0.068	0.946	$1.12\times 10^{-124}$	+23.54
PY	0.355	0.1808	0.038	0.843	0.376	0.1915	0.039	0.870	$1.05\times 10^{-9}$	+5.92
PC	0.476	0.2266	0.059	0.920	0.438	0.2152	0.049	0.910	$9.90\times 10^{-22}$	−7.98
Trait	ssGBLUP				ssGBLUPp				*p*-value	An increase/ reduction (%)
	$\overset{-}{X}$	SD	Min	Max	$\overset{-}{X}$	SD	Min	Max		
MY	0.367	0.1899	0.015	0.903	0.424	0.2123	0.017	0.927	$1.02\times 10^{-53}$	+15.53
FY	0.362	0.1838	0.005	0.899	0.421	0.2094	0.020	0.927	$4.09\times 10^{-59}$	+16.30
FC	0.443	0.2136	0.060	0.928	0.532	0.2495	0.055	0.957	$1.46\times 10^{-99}$	+20.09
PY	0.376	0.1942	0.007	0.907	0.403	0.2039	0.006	0.921	$3.31\times 10^{-13}$	+7.18
PC	0.483	0.2298	0.057	0.941	0.454	0.2150	0.026	0.936	$9.40\times 10^{-13}$	−6.00
Trait	MBLUP				MBLUPp				*p*-value	An increase/ reduction (%)
	$\overset{-}{X}$	SD	Min	Max	$\overset{-}{X}$	SD	Min	Max		
MY	0.419	0.2051	0.049	0.889	0.470	0.2257	0.057	0.924	$6.82\times 10^{-38}$	+12.17
FY	0.419	0.2050	0.049	0.889	0.475	0.2274	0.058	0.926	$2.26\times 10^{-44}$	+13.37
FC	0.442	0.2136	0.053	0.902	0.522	0.2454	0.067	0.942	$3.00\times 10^{-79}$	+18.10
PY	0.399	0.1971	0.046	0.874	0.466	0.2242	0.056	0.923	$5.61\times 10^{-67}$	+16.79
PC	0.480	0.2278	0.060	0.920	0.456	0.2204	0.054	0.913	$5.40\times 10^{-9}$	−5.00

BLUP—univariate, first lactation only, BLUPp—univariate, all lactations, ssGBLUP—univariate, first lactation only, ssGBLUPp—univariate, all lactations, MBLUP—multivariate, first lactation only, MBLUPp—multivariate, all lactations, $\overset{-}{X}$—mean, SD—standard deviation, Min—smallest value, Max—highest value, MY, kg—milk yield, FY, kg—milk fat yield, FC, % —milk fat content, PY, kg—milk protein yield, PC, %—milk protein content.

**Table 3 animals-13-00597-t003:** Descriptive statistics and p-values obtained using a two-tailed *t*-test for breeding values calculated using the BLUP(p), ssGBLUP(p) and MBLUP(p) models.

Trait	BLUP				BLUPp				*p*-Value
	$\overset{-}{X}$	SD	Min	Max	$\overset{-}{X}$	SD	Min	Max	
MY	45.6	206.57	−833.2	973.2	52.6	274.44	−1277	1140	0.116
FY	2.66	10.873	−39.9	116.1	2.68	14.321	−65.0	100.3	0.936
FC	0.010	0.1248	−0.549	1.40	0.009	0.1883	−0.853	1.74	0.817
PY	1.36	7.361	−35.8	67.4	1.58	8.117	−45.5	51.0	0.126
PC	−0.004	0.0602	−0.272	0.950	−0.001	0.0521	−0.242	0.842	0.003
Trait	ssGBLUP				ssGBLUPp				*p*-value
	$\overset{-}{X}$	SD	Min	Max	$\overset{-}{X}$	SD	Min	Max	
MY	−22.1	250.74	−1155	1029	30.7	351.91	−1530	1574	$2.40\times 10^{-21}$
FY	−1.03	11.621	−56.8	129.0	0.848	15.4857	−85.8	119.4	$4.37\times 10^{-14}$
FC	−0.003	0.1278	−0.576	1.393	−0.007	0.1887	−0.849	1.67	0.151
PY	−0.667	8.3835	−45.5	68.5	0.562	9.3995	−50.8	51.9	$3.57\times 10^{-14}$
PC	0.001	0.0614	−0.263	0.894	−0.0025	0.05621	−0.255	0.772	$2.72\times 10^{-4}$
Trait	MBLUP				MBLUPp				*p*-value
	$\overset{-}{X}$	SD	Min	Max	$\overset{-}{X}$	SD	Min	Max	
MY	48.2	304.56	−1199	2335	63.5	418.04	−2179	1690	0.022
FY	1.87	11.757	−46.2	90.4	3.38	19.656	−94.3	134.1	$2.84\times 10^{-7}$
FC	0.004	0.1227	−0.563	0.779	0.011	0.1806	−0.856	1.08	0.022
PY	0.806	7.9649	−38.0	115.7	2.05	12.520	−60.6	80.2	$1.03\times 10^{-10}$
PC	−0.003	0.0602	−0.285	1.08	−0.0002	0.05424	−0.199	0.720	0.002

BLUP—univariate, first lactation only, BLUPp—univariate, all lactations, ssGBLUP—univariate, first lactation only, ssGBLUPp—univariate, all lactations, MBLUP—multivariate, first lactation only, MBLUPp—multivariate, all lactations, $\overset{-}{X}$—mean, SD—standard deviation, Min—smallest value, Max—highest value, MY, kg—milk yield, FY, kg—milk fat yield, FC, % —milk fat content, PY, kg—milk protein yield, PC, %—milk protein content.

**Table 4 animals-13-00597-t004:** Descriptive statistics and *p*-values obtained using a two-tailed *t*-test for the accuracy of prediction of breeding values calculated using the BLUP(p), GBLUP(p) and ssGBLUP(p) models.

Trait	BLUP				GBLUP				ssGBLUP				Increase * (%)
	$\overset{-}{X}$	SD	Min	Max	$\overset{-}{X}$	SD	Min	Max	$\overset{-}{X}$	SD	Min	Max	
MY	0.496	0.0390	0.291	0.580	0.523	0.0521	0.050	0.618	0.545	0.0397	0.342	0.631	+9.88
FY	0.517	0.0379	0.304	0.600	0.521	0.0514	0.038	0.617	0.534	0.0406	0.334	0.623	+3.29
FC	0.630	0.0308	0.375	0.703	0.632	0.0468	0.149	0.712	0.650	0.0307	0.432	0.718	+3.17
PY	0.526	0.0374	0.310	0.608	0.545	0.0498	0.119	0.635	0.559	0.0384	0.354	0.643	+6.27
PC	0.697	0.0269	0.411	0.760	0.676	0.0459	0.196	0.754	0.714	0.0257	0.482	0.777	+2.44
Trait	BLUPp				GBLUPp				ssGBLUPp				Increase * (%)
	$\overset{-}{X}$	SD	Min	Max	$\overset{-}{X}$	SD	Min	Max	$\overset{-}{X}$	SD	Min	Max	
MY	0.554	0.0576	0.376	0.674	0.561	0.0534	0.384	0.704	0.615	0.0533	0.424	0.737	+11.01
FY	0.579	0.0596	0.403	0.704	0.561	0.0540	0.383	0.705	0.608	0.0543	0.414	0.733	+5.01
FC	0.753	0.0573	0.587	0.875	0.736	0.0509	0.572	0.862	0.766	0.0483	0.639	0.872	+1.73
PY	0.540	0.0593	0.357	0.644	0.529	0.0559	0.329	0.675	0.582	0.0559	0.376	0.709	+7.78
PC	0.624	0.0615	0.456	0.752	0.527	0.0605	0.316	0.682	0.649	0.0534	0.470	0.772	+4.01

BLUP—univariate, first lactation only, GBLUP—univariate, first lactation only, all genotyped, ssGBLUP—univariate, first lactation only, not all genotyped, * Percentage increase in accuracy obtained by the ssGBLUP model compared to the BLUP model, BLUPp—univariate, all lactations, GBLUPp—univariate, all lactations, all genotyped, ssGBLUPp—univariate, all lactations, not all genotyped, * Percentage increase in accuracy obtained by the ssGBLUPp model compared to the BLUPpe model, $\overset{-}{X}$—mean, SD—standard deviation, Min—smallest value, Max—highest value, MY, kg—milk yield, FY, kg—milk fat yield, FC, % —milk fat content, PY, kg—milk protein yield, PC, %—milk protein content.

**Table 5 animals-13-00597-t005:** Descriptive statistics for breeding values calculated using the BLUP(p), GBLUP(p) and ssGBLUP(p) models.

Trait	BLUP				GBLUP				ssGBLUP			
	$\overset{-}{X}$	SD	Min	Max	$\overset{-}{X}$	SD	Min	Max	$\overset{-}{X}$	SD	Min	Max
MY	117.2	268.25	−728.4	973.2	$4.62\times 10^{-8}$	349.241	−1219.3	880.3	0.120	346.6788	−1089.1	1029.0
FY	6.50	14.406	−30.3	116.1	$4.74\times 10^{-9}$	18.063	−59.9	159.4	0.006	16.4807	−51.9	129.0
FC	0.023	0.172	−0.505	1.40	$6.80\times 10^{-12}$	0.193	−0.580	1.67	$2.38\times 10^{-5}$	0.183	−0.576	1.39
PY	3.446	9.4380	−29.1	67.4	$-1.33\times 10^{-9}$	11.70	−43.5	77.7	0.004	11.3563	−36.3	68.5
PC	−0.012	0.0773	−0.253	0.950	$2.46\times 10^{-11}$	0.079	−0.251	0.940	$-1.03\times 10^{-5}$	0.082	−0.258	0.894
Trait	BLUPp				GBLUPp				ssGBLUPp			
	$\overset{-}{X}$	SD	Min	Max	$\overset{-}{X}$	SD	Min	Max	$\overset{-}{X}$	SD	Min	Max
MY	131.9	337.34	−1052	1015	$-8.50\times 10^{-8}$	415.623	−1468.4	1258.3	−0.470	458.5280	−1482.1	1377.8
FY	6.45	18.273	−65.3	100.3	$-5.09\times 10^{-10}$	19.626	−75.5	136.6	−0.011	20.8048	−85.8	119.4
FC	0.023	0.2531	−0.732	1.74	$5.11\times 10^{-11}$	0.245	−0.809	1.87	$1.51\times 10^{-4}$	0.261	−0.682	1.67
PY	4.23	10.0231	−41.2	51.0	$-5.74\times 10^{-9}$	11.32	−39.4	51.0	−0.011	12.3842	−49.3	51.9
PC	0.001	0.0666	−0.197	0.842	$5.26\times 10^{-11}$	0.056	−0.189	0.557	$7.38\times 10^{-6}$	0.075	−0.234	0.772

BLUPp—univariate, all lactations, GBLUPp—univariate, first lactation only, all genotyped, ssGBLUPp—univariate, all lactations, not all genotyped, $\overset{-}{X}$—mean, SD—standard deviation, Min—smallest value, Max—highest value, MY, kg—milk yield, FY, kg—milk fat yield, FC, % —milk fat content, PY, kg—milk protein yield, PC, %—milk protein content.

**Table 6 animals-13-00597-t006:** Spearman’s rank (above diagonal) and Pearson’s correlation coefficients (below diagonal) between predicted breeding values for milk production traits in first lactation.

MY	BLUP	GBLUP	ssGBLUP
BLUP	1.000	0.785	0.842
GBLUP	0.789	1.000	0.935
ssGBLUP	0.849	0.937	1.000
FY	BLUP	GBLUP	ssGBLUP
BLUP	1.000	0.803	0.849
GBLUP	0.822	1.000	0.953
ssGBLUP	0.862	0.958	1.000
FC	BLUP	GBLUP	ssGBLUP
BLUP	1.000	0.820	0.879
GBLUP	0.850	1.000	0.952
ssGBLUP	0.899	0.958	1.000
PY	BLUP	GBLUP	ssGBLUP
BLUP	1.000	0.787	0.848
GBLUP	0.804	1.000	0.935
ssGBLUP	0.864	0.938	1.000
PC	BLUP	GBLUP	ssGBLUP
BLUP	1.000	0.856	0.910
GBLUP	0.881	1.000	0.956
ssGBLUP	0.924	0.962	1.000

BLUP—univariate, first lactation only, GBLUP—univariate, first lactation only, ssGBLUP—univariate, first lactation only, MY, kg—milk yield, FY, kg—milk fat yield, FC, % —milk fat content, PY, kg—milk protein yield, PC, %—milk protein content.

**Table 7 animals-13-00597-t007:** Spearman’s rank (above diagonal) and Pearson’s correlation coefficients (below diagonal) between predicted breeding values for milk production traits in all lactations.

MY	BLUPp	GBLUPp	ssGBLUPp
BLUPp	1.000	0.723	0.802
GBLUPp	0.733	1.000	0.925
ssGBLUPp	0.811	0.935	1.000
FY	BLUPp	GBLUPp	ssGBLUPp
BLUPp	1.000	0.761	0.841
GBLUPp	0.783	1.000	0.941
ssGBLUPp	0.859	0.945	1.000
FC	BLUPp	GBLUPp	ssGBLUPp
BLUPp	1.000	0.881	0.932
GBLUPp	0.902	1.000	0.959
ssGBLUPp	0.946	0.962	1.000
PY	BLUPp	GBLUPp	ssGBLUPp
BLUPp	1.000	0.702	0.779
GBLUPp	0.732	1.000	0.921
ssGBLUPp	0.807	0.933	1.000
PC	BLUPp	GBLUPp	ssGBLUPp
BLUPp	1.000	0.792	0.867
GBLUPp	0.848	1.000	0.945
ssGBLUPp	0.905	0.955	1.000

BLUPp—univariate, all lactations, GBLUPp—univariate, all lactations, ssGBLUPp—univariate, all lactations, MY, kg—milk yield, FY, kg—milk fat yield, FC, % —milk fat content, PY, kg—milk protein yield, PC, %—milk protein content.

## Data Availability

The original data used in this article are available by contacting the corresponding author upon request. After acceptance for publication they will be available on the project website database: https://bioitgenoselect.polj.uns.ac.rs/genoselect/data-p1, accessed on 12 December, 2022.

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
