# Peer review of "Mathematical Modeling and Software Tools for Breeding Value Estimation Based on Phenotypic, Pedigree and Genomic Information of Holstein Friesian Cattle in Serbia"

_animals, 2023, doi:10.3390/ani13040597_

Round 1

Reviewer 1 Report

Major comments

-       You should include recent data about cattle breeding and cattle production sector in Serbia in the part of the ‘’introduction’’

-       you should discuss the economic impact of your results at national level

-       You should revise the conclusions: don’t report the aim of the study, you could reduce the length of this part

Minor comments

§  L155-156: remove the sentence

§  L496-502: add appropriate references 

§  L514-521: add appropriate references

Author Response

Major comments

-       You should include recent data about cattle breeding and cattle production sector in Serbia in the part of the ‘’introduction’’

Added a paragraph in the introduction. Lines 153 to 162.

-       You should discuss” the economic impact of your results at national level

Added last sentence in conclusion. Lines 738 to 741.

-       You should revise the conclusions: don’t report the aim of the study, you could reduce the length of this part

The sentence The aim of the stydy … has been removed from the conclusions.

Minor comments

§  L155-156: remove the sentence

The sentence has been removed. The proper sentence has been inserted. Numbers of tables were corrected throughout the text. Lines 215 to 216.

§  L496-502: add appropriate references

We have added references. Line 639.

§  L514-521: add appropriate references

We have added references. Line 659.

Reviewer 2 Report

The paper aims to present models for explaining the influence of genetic factors on important production characteristics of cattle breeding. With this goal, the authors dealt with the field of theoretical models and with the level of numerical processing. 

The second chapter presents eight mathematical models using the cited literature.

I believe that the literature stade-of-arts should be expanded. In particular, I missed citations of articles on the breeding of Holstein cows, in which other authors analyzed the effects of breeding success on the economic characteristics of the breeding.  Unfortunately, the authors are content with citing articles using any of the eight models. So instead of citing similar analyzes of Holstein cow breeding, readers of the article are only referring to Buffaloes [27] and Huacaya Alpaca [28]. A host of models evaluate the effects such as the success of genetic improvement, increasing quality of care, better feed composition, and seasonal effect milk production at the population level, cf. [Z1], [Z2]. 

I recommend incorporating the following citations into the text:

[Z1] Zavadilova, J.: Definition of subgroups for fixed regression in the test-day animal model for milk production of Holstein cattle in the Czech Republic. Czech J. Anim. Sci. 50, vol. 1 (2005), 7-13.

[Z2] Zavadilova, J.: Genetic parameters for a test-day model with random regressions for production traits of Czech Holstein cattle. Czech J. Anim. Sci. 50, vol. 4 (2005), 142-150.

Further deepening of the literature survey is necessary.

The third chapter is devoted to the numerical study. The authors focus on comparing the empirical values of selected production characteristics with their estimates, which are obtained using regression analysis. 

In the numerical part, as a reader, I would welcome the presentation of specific examples of input design matrices, covariances, and correlation coefficients. 

The presentation of empirical values and estimated values is much more interesting for the reader. Cf. Q3).

I see no point in presenting these differences, cf. Q5). If the tables aim to demonstrate the quality of the models used, the presentation of the table with the coefficients of determination is sufficient. 

This is the most compelling reason for the final evaluation of a peer-reviewed paper: major revision.

Specific questions and comments:

Q1) The paper does not mention how the 305 daily feeds were obtained. Were daily milk values for each day of the lactation cycle available for the cows? Was the estimate of 305 daily yields calculated from monthly measurements of the daily milk yield? What approximation function was used (Wood's)?

Q2) What is the main reason for dividing dairy cows into three groups according to calving months? Is the 305 daily milk yield in group III statistically significantly smaller compared to the other groups?

Q3) Additional information about the studied herd of cows may be of interest to the reader. Can the average values of calving interval, days open, service per pregnancy, the average number of lactations per dairy cow, etc. be added? This will allow other authors to compare their results with your study and may contribute to higher citations of your article.

Q4) Table 1: What are the units of the quantity MY? In Tab. 2 you probably state MY in kilograms/day. The correctness of the value in Tab. 1 needs to be verified.

In Tab. 2: Why is the value -22+/-250.74? This is not an estimate of the value but the difference between the actual value and the estimated value, i.e. = Y - hat{Y} ?

Q5) I see no point in presenting these differences, cf. Q4). If the tables aim to demonstrate the quality of the models used, the presentation of the table with the coefficients of determination is sufficient.  

The presentation of empirical values and estimated values is much more interesting for the reader. Cf. Q4). 

Q6) Statistical criteria can be used to compare the quality of models: residual sum of squares, modified index of determination, Akaike criterion. I consider the comparison of the accuracy of the models in Tab 1-4 to be unclear and tiring for the reader. I would welcome the presentation of the results similar to the tables in the article [Z1].

Q7) It is not clear what the min and max values refer to (hat{Y}, Y-hat{Y})?

Q8) Line 545, 546: Accuracy is ...0.657, ... What does it mean? What is meant is the standard deviation or coefficient of determination?

Q9) Line 344-347: It is necessary to separate this text out of this subsection into a separate subsection. It is necessary to formulate a null hypothesis and complete the citation of the test statistic. List test assumptions, test statistics, and list input data. 

Minor typos:

Line 155 Missing reference

Line 166 Add the "hat" symbol above cov, sigma, and rho. The symbol "hat" in statistics indicates a parameter estimate. Alternatively: use the symbol s instead of sigma (the usual designation of sample standard deviation), and use the symbol r instead of rho (r is the typical symbol of the sample correlation coefficient).

Line 172 What is FYS?  Is it three matrices or just one matrix?

Line 213 pe - > p_e ... it is more appropriate to use a subscript, or only the vector p, must be changed throughout the text

Line 214 a is a vector of UNKNOWN parameters,  b is a vector of UNKNOWN parameters, 

Line 241 Write all matrices in bold!

Line 335 Explain the tensor product symbol.

Line 371 ?preGSF90

Throughout the article: missing commas, dots after formulas

Author Response

The second chapter presents eight mathematical models using the cited literature. I believe that the literature state-of-arts should be expanded. In particular, I missed citations of articles on the breeding of Holstein cows, in which other authors analyzed the effects of breeding success on the economic characteristics of the breeding. 

Unfortunately, the authors are content with citing articles using any of the eight models. So instead of citing similar analyzes of Holstein cow breeding, readers of the article are only referring to Buffaloes [27] and Huacaya Alpaca [28]. A host of models evaluate the effects such as the success of genetic improvement, increasing quality of care, better feed composition, and seasonal effect milk production at the population level, cf. [Z1], [Z2]. I recommend incorporating the following citations into the text:

[Z1] Zavadilova, J.: Definition of subgroups for fixed regression in the test-day animal model for milk production of Holstein cattle in the Czech Republic. Czech J. Anim. Sci. 50, vol. 1 (2005), 7-13.

[Z2] Zavadilova, J.: Genetic parameters for a test-day model with random regressions for production traits of Czech Holstein cattle. Czech J. Anim. Sci. 50, vol. 4 (2005), 142-150.

Further deepening of the literature survey is necessary.

In accordance with the recommendation, these two papers were added as additional ones in the field of dairy cattle breeding. In accordance with the recommendation, we have added Z1 as well as additional ones in the area of dairy cattle. Z2 is related to the evaluation of genetic parameters, therefore we are not able to citation it in this research.

The third chapters are devoted to the numerical study. The authors focus on comparing the empirical values of selected production characteristics with their estimates, which are obtained using regression analysis. 

In the numerical part, as a reader, I would welcome the presentation of specific examples of input design matrices, covariances, and correlation coefficients. 

The estimation of the components of variances, heritabilities and correlations is the subject of previous research that should be published in another paper. In this paper, we focused only on breeding value estimates and their accuracy.

The presentation of empirical values and estimated values is much more interesting for the reader. Cf. Q3).

I see no point in presenting these differences, cf. Q5). If the tables aim to demonstrate the quality of the models used, the presentation of the table with the coefficients of determination is sufficient.

This is the most compelling reason for the final evaluation of a peer-reviewed paper: major revision.

As suggested by the review, we added results related to Spearman's and Pearson's correlation coefficients for model comparison (lines 607 to 636). Also, the accuracy parameter of the estimated breeding values was used for the quality of the model. If the predicted breeding values is more accurate, the model is better, and if there are statistically significant differences, it means that the choice of model has an impact on the obtained estimates.

Specific questions and comments:

Q1) The paper does not mention how the 305 daily feeds were obtained. Were daily milk values for each day of the lactation cycle available for the cows? Was the estimate of 305 daily yields calculated from monthly measurements of the daily milk yield? What approximation function was used (Wood's)?

One of the recommended methods according to ICAR was used: Methods for calculating daily yields from AM/PM milkings - Method of Delorenzo and Wiggans (1986). Lines 179 to 180.

Q2) What is the main reason for dividing dairy cows into three groups according to calving months? Is the 305 daily milk yield in group III statistically significantly smaller compared to the other groups?

The months were grouped in relation to weather conditions (temperature and humidity of the environment): first group - winter season; the second spring autumn, and the third summer. The influence of factors is also part of the research related to the assessment of genetic parameters; therefore the results of defining the model are not shown in this research. Already, the model that best fits the available data has been used to estimate breeding values and their accuracies.

Q3) Additional information about the studied herd of cows may be of interest to the reader. Can the average values of calving interval, days open, service per pregnancy, the average number of lactations per dairy cow, etc. be added? This will allow other authors to compare their results with your study and may contribute to higher citations of your article.

During this project, we focused only on milk production traits because this is the first research related to the use of genomic data in Serbia, so we reduced the number of traits. In some other research, we will definitely form a database with data and other economically important traits.

Q4) Table 2: What are the units of the quantity MY? In Tab. 2 you probably state MY in kilograms/day. The correctness of the value in Tab. 2 needs to be verified.

In the material it is written that milk yield characteristics refer to lactation production standardized to 305 days, not daily. Based on the suggestion, we have added an abbreviation to each table in which the unit is expressed (kg, %).

In Tab. 3: Why is the value -22+/-250.74? This is not an estimate of the value but the difference between the actual value and the estimated value, i.e. = Y - hat{Y} ?

According to suggestions, all tables 2, 3, 4 and 5 were modified as in the paper [Z1].

Q5) I see no point in presenting these differences, cf. Q4). If the tables aim to demonstrate the quality of the models used, the presentation of the table with the coefficients of determination is sufficient.  

The presentation of empirical values and estimated values is much more interesting for the reader. Cf. Q4). 

One of the criteria for comparing models is the accuracy of the estimates obtained. A model that gives more accurate estimates of breeding values is better for the corresponding data set.

Q6) Statistical criteria can be used to compare the quality of models: residual sum of squares, modified index of determination, Akaike criterion. I consider the comparison of the accuracy of the models in Tab 2-5 to be unclear and tiring for the reader. I would welcome the presentation of the results similar to the tables in the article [Z1].

In accordance with the recommendation, data for Spearman's and Pearson's correlation coefficients, as well as certain graphs, were added. Lines 607 to 636.

Q7) It is not clear what the min and max values refer to (hat{Y}, Y-hat{Y})?

Min and max refer to the values of estimated accuracy of breeding values or estimated breeding values for all considered animals depending on given model and MY, FY, FC, PY and PC.

Q8) Line 545, 546: Accuracy is ...0.657, ... What does it mean? What is meant is the standard deviation or coefficient of determination?

It is the accuracy of the breeding values prediction, defined in lines 266 and 296.

Q9) Line 344-347: It is necessary to separate this text out of this subsection into a separate subsection. It is necessary to formulate a null hypothesis and complete the citation of the test statistic. List test assumptions, test statistics, and list input data. 

According to the recommendation, the separate paragraphs 2.3 and 2.4, with hypothesis were created. Lines 406 to 447 . Also, as recommended, instead of Duncan's test, we presented the results for Spearman's and Pearson's correlation coefficient.

Minor typos:

Line 155 Missing reference

The proper sentence has been inserted. Numbers of tables were corrected throughout the text. Lines 215 to 216 .

Line 166 Add the "hat" symbol above cov, sigma, and rho. The symbol "hat" in statistics indicates a parameter estimate. Alternatively: use the symbol s instead of sigma (the usual designation of sample standard deviation), and use the symbol r instead of rho (r is the typical symbol of the sample correlation coefficient).

This has been incorporated in the text. Line 226.

Line 172 What is FYS?  Is it three matrices or just one matrix?

It is a single matrix containing the interaction between farm F, year Y and season S.

Line 213 pe - > p_e ... it is more appropriate to use a subscript, or only the vector p, must be changed throughout the text

pe is substituted with just p throughout the text.

Line 214 a is a vector of UNKNOWN parameters,  b is a vector of UNKNOWN parameters, 

It has been inserted in the text. Lines 274 to 275.

Line 241 Write all matrices in bold!

A, B are alleles, not matrix, if we understood you correctly.

Line 335 Explain the tensor product symbol.

The explanation has been inserted in the formula. Line 396.

Line 371 ?preGSF90

Renamed to PreGSF90 and explained in one sentence. Lines 471 to 473.

Throughout the article: missing commas, dots after formulas

Dots have been inserted after formulas where appropriate.

Reviewer 3 Report

This manuscript was perfectly written and the models were described well.

This manuscript can be accepted after minor revision.

1. Line 75: “data on DNA, RNA, amino acid sequences and related information” should be “genomic data and related information” because genomic data is more than DNA, RNA, amino acid sequences.

2. Line 85: “reference group” may be “reference (train set) group”.

Author Response

1. Line 75: “data on DNA, RNA, amino acid sequences and related information” should be “genomic data and related information” because genomic data is more than DNA, RNA, amino acid sequences.

This suggestion has been accepted and incorporated in the text line 75.

2. Line 85: “reference group” may be “reference (train set) group”.

This suggestion has been accepted and incorporated in the text line 85.

Round 2

Reviewer 2 Report

The authors creatively used almost all comments and suggestions from my first review,  improved the text of the paper, especially revised the tables, added new figures, and supplemented the bibliography.  The original tables that analyzed the differences between measured values and predictions have been removed.

The performed correlation analysis made it possible to better express the quality of the constructed regression models.

It is worth emphasizing the exceptionally clear response to the reviewers' comments and marking in red changes and newly added text of the paper.

The new version of the paper is suitable for publication in the Animals journal, but please consider a few minor remarks listed below.

Line 15.  only [the] first
Line 17.
 for [the] estimation
Line 26.  repeatibility/
repeatability and and reproducibility
Line 34 size of [the] reference population
Line 89 monly/moustly
Line 111 … cows.This / … cows. This
Line 126 refered/referred
Line 142 [is] related to
Line 174 [sample] correlation
Line 175 [sample] variances
Line 383 n is [a] number of observations
Line 416 used tools / using tools
Line 461 / Table 2, 2nd row, 9th row: \overline
{X}_    [cancel the dash symbol _ ] 
Line 483 / Table 3, 2nd row, 9th row: \overline{X}_    [cancel the dash symbol _ ] 
Line 505 / Table 4, 2nd row, 9th row: \overline{X}_    [cancel the dash symbol _ ] 
Line 523 / Table 5, 2nd row, 9th row: \overline{X}_    [cancel the dash symbol _ ] 
Line 567 nowadays, are / nowadays are
Line 576 easier data / more accessible data
Line 590 Observing data / While observing data
Line 617 was13/ was 13
Line 618 in our research the difference / in our research, the difference
Line 632 Comparison / A comparison

Author Response

The authors would like to thank Reviewer 2 for careful comments in relation to the revision 1 of this manuscript.

Response to the Reviewer 2 comments:

Line 15.  only [the] first

done
Line 17.  for [the] estimation

done
Line 26.  repeatibility/ repeatability and and reproducibility

done
Line 34 size of [the] reference population

done

Line 89 monly/moustly

done
Line 111 … cows.This / … cows. This

done
Line 126 refered/referred

done
Line 142 [is] related to

done
Line 174 [sample] correlation

done
Line 175 [sample] variances

done
Line 383 n is [a] number of observations

done
Line 416 used tools / using tools

done
Line 461 / Table 2, 2nd row, 9th row: \overline{X}_    [cancel the dashsymbol _ ]

done
Line 483 / Table 3, 2nd row, 9th row: \overline{X}_    [cancel the dashsymbol _ ]

done
Line 505 / Table 4, 2nd row, 9th row: \overline{X}_    [cancel the dashsymbol _ ]

done
Line 523 / Table 5, 2nd row, 9th row: \overline{X}_    [cancel the dashsymbol _ ]

done
Line 567 nowadays, are / nowadays are

done
Line 576 easier data / more accessible data

done
Line 590 Observing data / While observing data

done

Line 617 was13/ was 13

done
Line 618 in our research the difference / in our research, the difference

done
Line 632 Comparison / A comparison

done